# Dosimetric Optimization of a Laser-Driven Irradiation Facility Using the G4-ELIMED Application

Sergio Mingo Barba [1,2,*], Francesco Schillaci [3], Roberto Catalano [4,5], Giada Petringa [3,4], Daniele Margarone [3,6] and Giuseppe Antonio Pablo Cirrone [4,5]

1  School of Engineering, Zurich University of Applied Sciences, 8400 Winterthur, Switzerland
2  Chemistry Department, University of Fribourg, 1700 Fribourg, Switzerland
3  ELI–Beamlines Center, Institute of Physics, Czech Academy of Sciences, 252 41 Dolní Břežany, Czech Republic; francesco.schillaci@eli-beams.eu (F.S.); giada.petringa@lns.infn.it (G.P.); D.Margarone@qub.ac.uk (D.M.)
4  Laboratori Nazionali del SUD, Istituto Nazionale di Fisica Nucleare (LNS-INFN), 95125 Catania, Italy; catalano@lns.infn.it (R.C.); pablo.cirrone@infn.it (G.A.P.C.)
5  Physics and Astronomy Department "E Majorana", University of Catania, 95125 Catania, Italy
6  Centre for Plasma Physics, School of Mathematics and Physics, Queen's University of Belfast, Belfast BT7 1NN, UK
*  Correspondence: ming@zhaw.ch

**Abstract:** ELIMED has been developed and installed at ELI beamlines as a part of the ELIMAIA beamline to transport, monitor, and use laser-driven ion beams suitable for multidisciplinary applications, including biomedical ones. This paper aims to investigate the feasibility to perform radiobiological experiments using laser-accelerated proton beams with intermediate energies (up to 30 MeV). To reach this goal, we simulate a proton source based on experimental data like the ones expected to be available in the first phase of ELIMED commissioning by using the G4-ELIMED application (an application based on the Geant4 toolkit that simulates the full ELIMED beamline). This allows the study of transmission efficiency and the final characteristics of the proton beam at the sample irradiation point. The Energy Selector System is used as an active energy modulator to obtain the desired beam features in a relatively short irradiation time (around 6 min). Furthermore, we demonstrate the capability of the beamline to filter out other ion contaminants, typically co-accelerated in a laser-plasma environment. These results can be considered as a detailed feasibility study for the use of ELIMED for various user applications such as radiobiological experiments with ultrahigh dose rate proton beams.

**Keywords:** Monte Carlo simulations; Geant4; laser-accelerated ion beams

## 1. Introduction

High power laser-plasma interaction is a new and innovative approach to produce and accelerate particle beams [1]. The interaction of ultrahigh laser intensities ($>10^{19}$ W/cm$^2$) with a thin (~μm) solid target results in the generation of extremely high magnetic and electric fields that produce a plasma and relativistic electrons (known as "hot electrons") propagating into the vacuum and creating a quasi-static sheath electric field at the target-vacuum interface. Such a field ionizes the target rear side and accelerates the ions outwards. The characteristics of the laser-accelerated ion beam will depend on the used laser and target parameters.

A laser-plasma ion accelerator can be considered as a "multi-color" source where different kinds of ionizing radiations (protons/ions, gamma/X-rays, electrons, and neutrons) can be produced simultaneously. Additionally, such accelerators are expected to generate ultra-high dose rate beams, which are orders of magnitude higher than those currently being proposed for the "FLASH" radiotherapy approach [2,3]. Moreover, a laser-based approach could potentially reduce the overall size and cost of an accelerator installation.

In this framework, the ELIMAIA (ELI Multidisciplinary Applications of laser-Ion Acceleration) beamline [4] at the ELI Beamlines (Extreme Light Infrastructure) Centre aims to provide ion beams accelerated by high repetition-rate petawatt-class lasers suitable for multidisciplinary user applications. The two major subsystems of ELIMAIA are the Ion Accelerator and ELIMED (ELI Beamlines MEDical and multidisciplinary applications) sections [5,6]. ELIMED, in turn, consists of three main sub-sections: (i) the ion collection and focusing part, (ii) the ion energy selection, and (iii) the in-air transport section. The collection and focusing section aims to collimate the laser-accelerated ion beam and reduce its peculiar large divergence. This part is made of a set of five Permanent-Magnet Quadrupoles (PMQs). As described in [7], the PMQs have different lengths (one is 160 mm long, two are 120 mm, and the other two 80 mm) and a field gradient of around 100 T/m over a 36 mm magnetic bore. The PMQs are used to properly inject the accelerated particles downstream into the Energy Selection System (ESS). Hence, the full section from the target to the first collimator of the chicane is arranged in a way that the matching condition between collection and selection sections are respected. This means that the drift between quadrupoles is chosen to keep the emittance and Twiss Parameters within the required values of the chicane (Table 1). Also, the transport matrix conditions to have a waist on the horizontal axis and a parallel beam on the vertical axis are respected (it means M12 = 0 and M44 = 0). The ESS is a chicane made of four laminated resistive dipoles [8]. The technology used for the laminated yokes of the magnets (98% of packing factor) allows fast changes (1 Hz) in the magnetic field intensity based on the required ion species, ion energy, and energy bandwidth. This allows to change the selected ion energy between different shots, i.e., the ESS can be used as an active energy modulator. This is a unique feature of the ELIMED beamline not available at other laser-based accelerator facilities. Downstream of the ESS, a set of electromagnets (two electromagnetic quadrupoles and two steerers) are available to allow a final shaping of the particle beam and to correct for systematic misalignments prior to its final delivery onto the user sample in the in-air dosimetry end-station, which is separated from the in-vacuum section by a thin kapton window. A detailed description of the ELIMAIA beamline, along with the ELIMED transport magnetic elements can be found here [4,6,9].

**Table 1.** ESS acceptance parameters. The parameters are defined in [8] and summarized here.

|  | $X\theta_x$ | $Y\theta_y$ | XY |
|---|---|---|---|
| $\alpha$ | 0.8401 | 0.3556 | 0.0002 |
| $\beta$ (mm/$\pi$·mrad) | 2.7094 | 2.4484 | 0.9112 |
| Emit. Norm (rms) $\pi$ mm·mrad | 2.9506 | 3.9324 | 24.15 mm$^2$ |
| $X_{max}$ | $Y_{max}$ | $\theta_{x, max}$ | $\theta_{y, max}$ |
| 14.97 mm | 14.99 mm | 8.632 mrad | 7.162 mrad |

The ion and proton beams transported along ELIMED are characterized and monitored online in terms of energy, fluence, and spatial profile through a set of in-line detectors [10]. Diamond and silicon carbide detectors are extensively used in a Time-Of-Flight (TOF) configuration [11,12] to rapidly retrieve the beam energy spectra at different positions along the beamline. Furthermore, accurate shot-to-shot measurements of the dose released at the end of the beamline (where the user samples are placed) can be performed. ELIMED's absolute dosimetry systems are independent of the ultra-high dose rate (up to $10^9$ Gy/s) and allow to perform online absolute dose determination with an accuracy better than 5%, thus satisfying the internationally established clinical requirements [13–15]. The ELIMED dosimetry system is based on three main devices: (i) a Secondary Electron Monitoring (SEM), (ii) a Multi-Gap Ionization Chamber (MGIC), and (iii) a Faraday Cup (FC) for absolute dosimetry. Passive detectors, such as CR39 and radiochromic films (RCF), are also used to benchmark active ion diagnostic and dosimetry devices. The entire ELIMED beamline (considering the initial ion source as an input) can be fully simulated using the developed ELIMED application of the Geant4 Monte Carlo toolkit [16–18].

In this study, the capability of the G4-ELIMED application was exploited to optimize the transmission efficiency along the beamline and the dosimetric characteristics of the final beam. The beamline configuration was optimized to obtain a final beam suitable for pilot radiobiology experiments by using laser-generated proton beams centered around 20 MeV. A Spread-Out Bragg Peak (SOBP) was generated using the ESS as an active energy modulator, which was the main aim of this study and at the same time producing a depth-dose profile similar to the ones required to carry out radiobiological experiments. Additionally, the removal of unwanted ion species accelerated in the laser-generated plasma, such as carbon ions, was studied to assure the capability of the beamline to filter out ion beam contaminants that can be detrimental for accurate dosimetric studies.

## 2. Materials and Methods

### 2.1. The G4-ELIMED Application

A dedicated Monte Carlo application has been developed to simulate the full ELIMED beamline and, in particular, to assess the dosimetric features of the ion beam on the user sample [16]. The Geant4 (GEometry ANd Tracking) toolkit [19–21], version 10.03, was selected as the most appropriate code for the ELIMED transport and dosimetry beamline simulation for its robustness, versatility, and reliability of the implemented physical processes.

The G4-ELIMED application realistically reproduces each element of the beamline, both in terms of geometry and magnetic features; it includes the detectors for the beam diagnostics and dosimetry (e.g., the SEM detector and the Faraday cup) and allows to retrieve complementary key information, such as secondary radiation emission along the beam transport section, ion dose distributions at the irradiated sample, and many others.

Since the ELIMED beamline was designed and realized to work in a wide range of applications (e.g., radiation chemistry like pulsed radiolysis of water [22,23], nuclear physics for generation of isotopes for Positron Emission Tomography (PET) [24,25], cultural heritage using proton activation analysis (PAA) techniques [26], and material science through radiation stress-tests, including electronics for space application [27,28]), the beamline setup can be easily modified, thus the simulation tools should support the exploitation of such beamline modularity. The user-friendly interface of the code allows its simple use also by non-expert users.

In this work, all the simulations were carried out with $10^5$ initial particles and a maximum simulation step of 50 µm. These values were chosen to obtain high-quality results while maintaining a reasonable computational time.

### 2.2. Source Implementation

Since experimental data from the ELIMAIA-ELIMED source are not available yet, a realistic experimental source term (based on data from the J-KAREN-P PW-class laser facility in Japan [29]) was implemented in the simulation. The energy and angular distributions were based on the data from Dover et al. [30], where they used a laser beam with an irradiance of $5 \times 10^{21}$ W·cm$^{-2}$ and a stainless steel target with a thickness of 5 µm placed at $45^0$ with respect to the laser direction. On the other hand, the spatial distribution was assumed to be Gaussian with a standard deviation equal to 10 µm. All the distributions are presented in Figure 1.

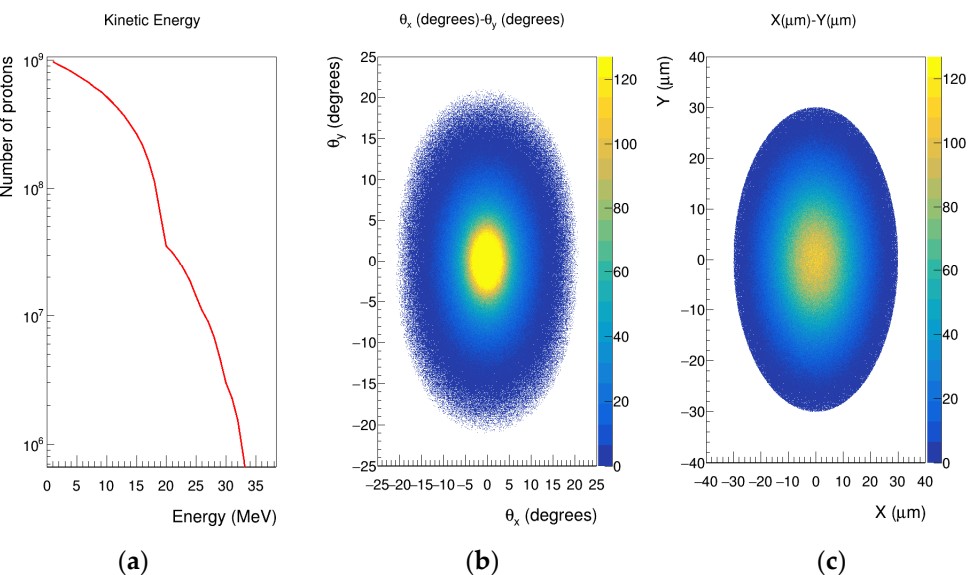

**Figure 1.** Distributions of the implemented source: (**a**) Initial kinetic energy distribution following the data presented in Dover et al. [30]; (**b**) Initial angular distribution. Protons at different energies have a maximum half-angle between 2.4 (highest energy) and 21 (lowest energy) degrees according to the data presented in Dover et al. [30]; (**c**) Initial position of the protons in the XY plane.

*2.3. Depth-Dose Profile Generation*

Clinical irradiations with ion beams are usually carried out using a Spread-Out Bragg Peak (or SOBP), i.e., a flat depth dose distribution is needed to uniformly irradiate a solid tumor.

The SOBP is the result of many beams of different energies and intensities added up with an appropriate weighting function. Usually, the energy change is done by some passive energy modulation system (e.g., a wheel modulator [31] or a ridge filter [32]) coupled with additional range shifters. These components are not required along the ELIMED beamline because the initial energy spectrum of laser-accelerated beams is intrinsically poly-energetic and the ESS can be used as an active energy modulator.

The only problem with such an approach is that the configuration of the focusing system needs to be changed to properly inject protons at different kinetic energies into the ESS. This will increase the time required to perform a certain experiment. For this reason, only the configurations to focus four different energies (18, 20, 22, and 25 MeV) were considered in this work. These changes have been calculated keeping the same sequence of magnets and limiting the displacement as much as possible in order to reduce the time necessary for repositioning the magnets. In such a way the transmission efficiency is not optimal, but the full irradiation time is limited to about 6 min. On the other hand, four setups calculated to maximize the transmission efficiency would require changing the sequence of the magnets, which means several hours because this operation cannot be performed under vacuum and involves the manipulation of heavy objects (the weight of the smaller quadrupole is about 70 kg).

In the simulated configurations, the first and the fourth PMQs have a length equal to 120 mm and positive polarity, while the second and the third PMQs have a length of 80 mm and negative polarity. The initial positions of the PMQs to transport the different energies are summarized in Table 2 and the relative distances between elements (D1, D2, D3, and D4) are clarified in Figure 2 where the generic scheme of the setup is presented.

**Table 2.** Initial positions of the four PMQs for the configurations used to focus different energy beams. The distances are calculated with respect to the position of the source.

| Energy (MeV) | D1 (mm) | D2 (mm) | D3 (mm) | D4 (mm) |
|---|---|---|---|---|
| 18 | 56.8 | 218.4 | 772.0 | 901.1 |
| 20 | 60.2 | 228.3 | 798.2 | 924.0 |
| 22 | 59.4 | 253.1 | 836.2 | 956.2 |
| 25 | 61.5 | 281.7 | 885.5 | 1005.5 |

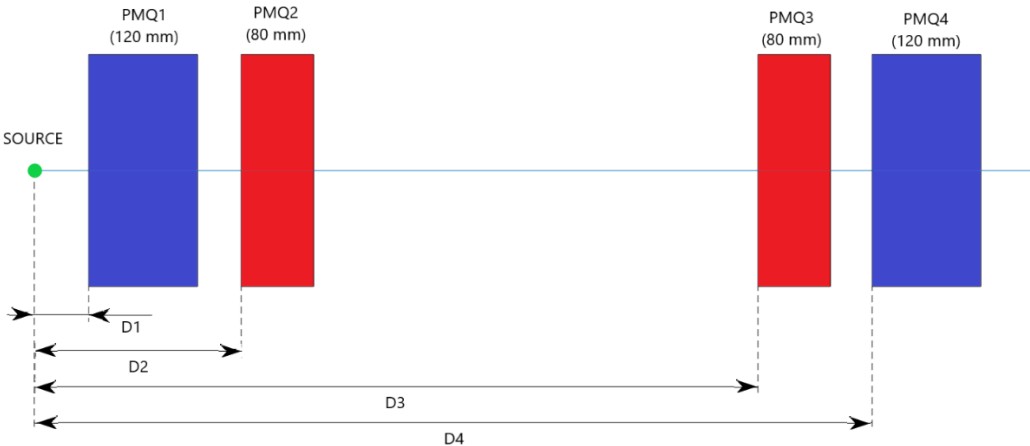

**Figure 2.** Layout of the collection and focusing section configuration. Values of the initial relative positions of the quadrupoles (D1, D2, D3, and D4) per each energy configuration are given in Table 2.

As mentioned above, the ESS can be used as an active energy modulator. This is realized by changing the magnetic field, i.e., the current intensity, of the dipoles to select different energies at different shots. Herein, the magnetic field values used in the energy selector are 0.243 T, 0.257 T, 0.269 T, and 0.287 T which are the necessary parameters to obtain protons with energies of 18 MeV, 20 MeV, 22 MeV, and 25 MeV, respectively, streaming on the reference trajectory. On the other hand, both the energy spread and the transmission efficiency depend on the aperture of the slit placed in the center of the ESS to select ions at different kinetic energies. Thus, a slit aperture equal to 30 mm was used to increase the transmission of protons, except in the 25 MeV case where a 20 mm slit aperture was used to reduce the energy spread and, hence, the distal fall-off of the SOBP.

*2.4. In-Air Configuration of the Beamline*

The detectors used for the online beam diagnostic and dosimetry were included in the simulations to consider the effect that they may introduce into the beam transport section. A brass scattering foil with a radius of 3 cm and a tantalum in-air collimator with 1.75 and 25 cm of inner and outer radius respectively were added to improve the characteristics of the final proton beam. To produce a proper SOBP, the thickness of the scattering foil (used to improve the lateral profiles) was varied between 200 and 320 μm. Furthermore, the in-air section length was decreased from 200 to 33 cm to reduce the scattering with the air and the loss of protons, which were important because of the low proton kinetic energies. A schematic layout of the in-air part configuration is presented in Figure 3.

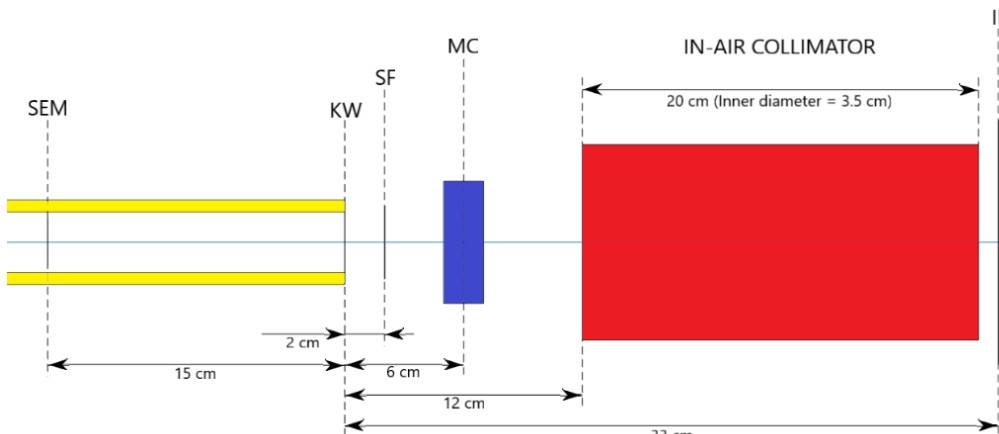

**Figure 3.** Layout of the in-air part of the beamline. The relative distances of the Secondary Electron Monitor (SEM), the Scattering Foil (SF), the Monitor Chamber (MC), the In-air Collimator, and the Irradiation Point (IP) are referred with respect to the Kapton Window (KW).

## 3. Results

Medical applications demand a high control of the beam characteristics. In this section, several clinically accepted parameters connected to the beam quality [33] were studied at the irradiation point to verify the capability of using the beamline to perform radiobiological experiments. The capability of the beamline to filter out unwanted carbon ions was additionally studied and discussed.

### 3.1. Lateral Profiles

The lateral profiles represent the relative dose distributions measured along the transversal axes with respect to the proton beam direction (in our case, the X and Y axes). To irradiate cells, flat distributions with very sharp lateral penumbras are desirable to ensure a homogeneous irradiation over all the cells. In this study, to reduce the impact of the noise produced by a lack of statistics in the quality parameters, the lateral profiles were normalized to the average value of the signal. The obtained lateral profiles are shown in Figure 4 and the corresponding beam quality parameters are summarized in Table 3.

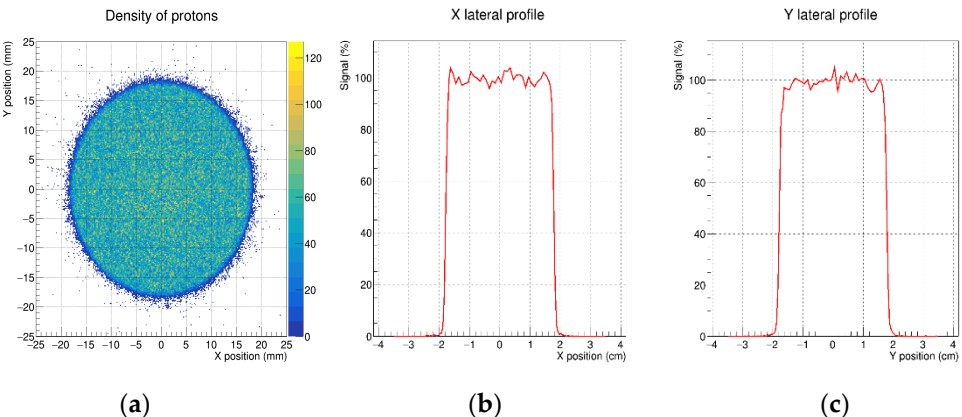

**Figure 4.** Distributions related with the lateral profiles at the irradiation point: (**a**) Density of protons in the XY plane produced by the combination of different energy beams; (**b**) Normalized lateral dose profiles on the X-axis obtained by the combination of different energy beams at the irradiation point; (**c**) Normalized lateral dose profiles on the Y-axis obtained by the combination of different energy beams at the irradiation point.

**Table 3.** Beam quality parameter tolerances and obtained results of the final lateral dose profiles. The calculation of quality parameters is defined in detail in [34,35].

| Parameter | Tolerance | X Profile | Y Profile |
|---|---|---|---|
| FWHM | As close as possible to the beam diameter | 3.57 cm | 3.56 cm |
| Left penumbra | ≤1.5 mm | 0.85 mm | 0.91 mm |
| Right penumbra | ≤1.5 mm | 0.82 mm | 0.96 mm |
| Ratio 90%/50% | >0.9 | 0.96 | 0.95 |
| Flatness | ≤3% | 4.1% | 4.7% |
| Symmetry | 97–103% | 101.2% | 100.1% |

As observed in Table 3, the only parameter which was not falling within the required tolerances was the flatness. But, as it is observed in Figure 4, the profiles had a certain noise which was probably generated by a lack of statistics in the simulated proton histories.

### 3.2. Depth-Dose Profile

The depth-dose profile represents the dose deposited along the beam direction (in our case, the Z-axis). Herein, as previously mentioned, we tried to reproduce a SOBP with the combination of four different beam energies. The values of the depth-dose profile quality parameters are presented in Table 4 and the contribution of each energy together with their final combination are shown in Figure 5.

**Table 4.** Beam quality parameter tolerances and obtained results of the final depth-dose profile. The calculation of quality parameters is defined in detail in [31]. Here, the distal fall-off was defined as the 80–20% Penumbra.

| Parameter | Tolerance | Result |
|---|---|---|
| $M_{95}$ | ≥1 mm | 1.96 mm |
| Distal fall-off | <1.5 mm | 0.94 mm |
| Flatness | <5% | 3.6% |

**Figure 5.** Depth-dose profile obtained at the irradiation point with the contribution of the four selected energies: 18 (yellow line), 20 (magenta line), 22 (black line), and 25 (blue line) MeV; while the red line corresponds to the combination of all them.

In this case, all the beam quality parameters were within the tolerances recommended of the international dosimetry code of practice [33], therefore this depth-dose distribution would be acceptable for clinically relevant radiobiology irradiation.

### 3.3. Transmission Efficiency

Another important aspect to be discussed is the capability of the beamline to efficiently transport protons around a given energy. The transmission efficiency was defined as the percentage of transmitted protons within ±10% of the selected kinetic energy (e.g, in the 20 MeV case, we would consider protons with kinetic energy between 18 and 22 MeV). The transmission efficiencies at diverse positions along the beamline are compiled in Table 5.

**Table 5.** Transmission efficiency values for different energy configurations and at different points along the ELIMED beamline.

| Energy (MeV) | After PMQs (%) | After ESS (%) | After Kapton Window (%) | Final (%) |
|---|---|---|---|---|
| 18 | 39 | 2.93 | 2.67 | 0.37 |
| 20 | 42.7 | 6.38 | 5.77 | 0.91 |
| 22 | 37.1 | 6.56 | 6.2 | 0.81 |
| 25 | 34.4 | 6.31 | 5.98 | 0.75 |

Despite the relatively low transmission efficiency for the given source term [30], it is important to stress that the low transmission values reported in Table 5 are still acceptable thanks to the relatively large proton number at the source. Ultimately, considering the available laser repetition rate, we focus our study on the time required to perform a sample irradiation experiment, i.e., the time needed to deliver the required dose at the irradiation point.

In radiobiology experiments, doses of the order of 1–2 Gy are typically required. Hence, the number of shots necessary to reach a 2 Gy dose level was calculated considering $10^{10}$ initial protons in the full energy spectrum (i.e., between 0 and 33 MeV) and the transmission efficiency of every single energy. A total number of 2812 shots were obtained. Considering that the L3 HAPLS laser system at ELI Beamlines [36] can deliver PW-class laser pulses at 10 Hz, approximately 280 s will be needed to reach the required dose on the user sample. However, this does not consider the time required to change the position of the PMQs, which was calculated considering a speed of 1 mm/s, thus returning about 110 s. Ultimately, the whole irradiation time is estimated to be approximately 6 min for the given source term.

### 3.4. Transmission of Unwanted Ion Species

In all the simulations presented above, only protons were considered. However, depending on the specific target used to produce the proton beam, some heavier, high-energy ions may be generated. The presence of such unwanted ion species could be detrimental for the sample irradiation if they are not properly filtered out by means of the beamline elements, thus affecting the results of the experiment. Therefore, the heavy-ion transmission effects must be studied prior to the proton beam irradiation. The transmissions of carbon ions with different charge states (from $C^{1+}$ to $C^{6+}$) were considered at this stage. Such simulations were performed using a Carbon ion source similar to the proton one, but with a maximum cut-off energy calculated using the following empirical formula:

$$E_{cut-off}^{n+} = E_{cut-off}^{p} \cdot n/2, \tag{1}$$

where $E_{cut-off}^{n+}$ and $E_{cut-off}^{p}$ = 33 MeV are the maximum cut-off energies for carbon ions with a charge state equal to $n$ and for protons, respectively.

The simulations were carried out only up to the exit of the ESS where the $C^{1+}$, $C^{2+}$, and $C^{3+}$ beams were filtered out, while the $C^{4+}$, $C^{5+}$, and $C^{6+}$ beams presented a transmission over the total initial number of carbon ions of $10^{-2}$%, $3 \times 10^{-2}$% and $5 \times 10^{-2}$%, respectively. The simulations could be extended to the whole beamline but, as it is shown in Figure 6, the maximum kinetic energy after the ESS was around 70 MeV for $C^{5+}$ ions, and the maximum range in the air for these ions is around 21 cm (according to ICRU range tables [37]). So,

these ions would never be able to reach the irradiation point. Therefore, we can conclude that carbon ions in the given energy range are filtered out in the ELIMED beamline.

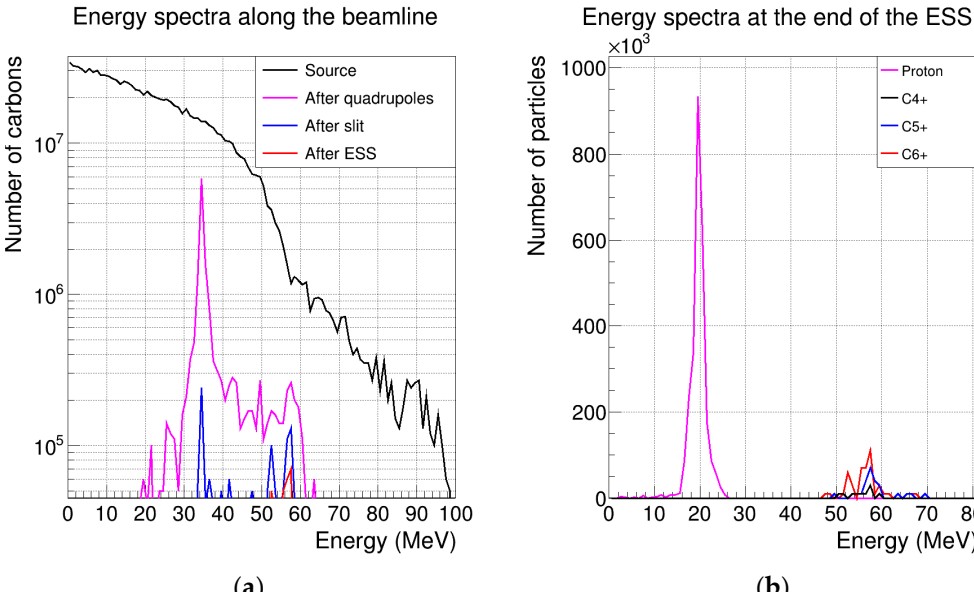

**Figure 6.** Energy spectra of carbon ions along ELIMED beamline: (**a**) Absolute energy spectra of $C^{6+}$ ions at different points along the ELIMED beamline; (**b**) Absolute energy spectra of protons (magenta line), $C^{4+}$ (black line), $C^{5+}$ (blue line) and $C^{6+}$ (red line) ions after the ESS. The spectra of $C^{1+}$, $C^{2+}$, and $C^{3+}$ beams are not included because they are fully filtered before the end of the ESS.

## 4. Discussion

The presented simulations used a realistic (experimental) high-power laser-accelerated proton source term as input for modeling its selection and transport. A relatively low kinetic energy window (centered at 20 MeV) was considered of interest for the pilot, in-vitro radiobiology experiments at the ELIMAIA-ELIMED beamline. The use of the Energy Selection System (ESS) as an active energy modulator was proven to be feasible, but it showed the drawback related to the need to re-positioning the four PMQs to transport protons with different kinetic energies. However, these changes in the configuration of the collecting system (PMQs) are not expected to drastically increase the overall irradiation time (approximately 2 additional minutes), thus it would enable radiobiological irradiations in a reasonable amount of time (around 6 min). Moreover, it is expected that at higher energies (around 60 MeV, i.e., already laying in the clinical window) a single configuration of the beam focusing system would allow a more efficient transport and injection into the ESS, both for low and high proton energies based on the presence of given resonances [38], thus potentially shortening the overall sample irradiation time required to create a clinical SOBP. Finally, it is noteworthy that the experimental source term (laser-driven proton source) can be improved in terms of total proton flux, thus enhancing the final dose delivered onto the user sample.

The numerical results presented are promising in terms of final particle beam properties and demonstrate to fulfill the quality requirements for clinical applications. Furthermore, it was shown that unwanted plasma ion species, such as carbon ions, are properly filtered out in the ELIMED beamline. Thus, once the experimental characterization of the proton source at ELIMAIA will be carried out, the ELIMED beamline can be fine-tuned based on the actual initial proton beam spectral and spatial features at the source, and ultimately be optimized for pilot radiobiological tests with ultrahigh dose-rate, ultrashort laser-accelerated beams.



**Author Contributions:** The paper's initial idea was proposed by D.M., G.A.P.C., G.P. and F.S. The conceptualization is by D.M., G.A.P.C., G.P., F.S. and S.M.B.; the software work and analysis were performed by S.M.B.; Supervision by R.C. and G.P. The original draft preparation was by S.M.B., G.A.P.C. and F.S. and the review and editing were performed by S.M.B., D.M., G.A.P.C. and F.S. All authors have read and agreed to the published version of the manuscript.

**Funding:** This research was funded by INFN in the framework of the Interdisciplinary Committee and by the MC-INFN and ELIMED projects; it was also funded by the Ministry of Education, Youth, and Sports of the Czech Republic through the project "Advanced Research Using High-Intensity Laser-Produced Photons and Particles" (CZ.02.1.010.00.016_0190000789).

**Data Availability Statement:** Data are available from the corresponding authors upon reasonable request.

**Conflicts of Interest:** The authors declare no conflict of interest.

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
