# Peer review of "Dosimetric Optimization of a Laser-Driven Irradiation Facility Using the G4-ELIMED Application"

_applsci, doi:10.3390/app11219823_

Round 1

Reviewer 1 Report

Report: Manuscript ID: applsci-1398504

Authors: Sergio Mingo Barba * , Francesco Schillaci , Roberto Catalano , Giada Petringa , Daniele Margarone , Giuseppe Antonio Pablo Cirrone

 The article by Barba et al. reports brief description of “ELI Multidisciplinary Applications of laser-Ion Acceleration (ELIMAIA)” and “ELIMED” end-station which is being developed at ELI Beamlines mainly planned for bio-medical application. The end station is designed to collect multi species ion beam from a PW laser driven laser ion accelerator and process the ion beam for user application. The detailed description of ELIMAIA and ELIMED are reported elsewhere. However, the authors reported an realistic approach to estimate the beam quality at the application point (irradiation) after transportation through the ELI-MED consisting of 4 quadruple magnets for collection and focusing of the ion beams and an “in-air” part of the endstation. Details of these magnets are not provided though. The details of the “in-air” part and their material compositions are also missing. The authors have used Geant4 toolkit, commonly used for passage of particles through matter. However, details of the simulation parameters used are missing. The end results are however convincing and the ion beam parameter at the application point seems usable for user experiments. Although substantial losses have been noticed during the transportation and shaping. Since, the ion beam source is still under construction, and ion source quality depends on several other parameter (target thickness, contrast of the laser etc), the final source parameter may be different from J-KAREN-P PW laser delivered ion source.

Finally, I recommend this article for publication after minor modifications. I also recommend to include few lines about the novelty of the approach in the manuscript.

Author Response

We thank the Referee for the very useful remarks that allowed to improve the overall quality of the manuscript.

Point 1: The detailed description of ELIMAIA and ELIMED are reported elsewhere. However, the authors reported an realistic approach to estimate the beam quality at the application point (irradiation) after transportation through the ELI-MED consisting of 4 quadruple magnets for collection and focusing of the ion beams and an “in-air” part of the endstation. Details of these magnets are not provided though.

Response 1: The details of the ion beam elements are presented in detail in previous papers cited in the manuscript. However, based on the Referee’s remark we decided to add a few lines to provide some brief information about the quadrupoles in the new version of the manuscript (lines 57-66).

Point 2: The details of the “in-air” part and their material compositions are also missing.

Response 2: We added additional information (material and dimensions) regarding the scattering foil and the in-air collimator (lines 188 and 189).

Point 3: The authors have used Geant4 toolkit, commonly used for passage of particles through matter. However, details of the simulation parameters used are missing.

Response 3: The G4-ELIMED application has a user-friendly interface. This means that, with different commands, we can easily change the configuration of the beamline, e.g., the ESS field intensity can be changed by simply choosing the energy that we want to select. However, some extra information about the simulation is now provided in lines 126-128 and 182-184.

Point 4: I also recommend to include few lines about the novelty of the approach in the manuscript.

Response 4: We included a few sentences to remark the novelty of the ELIMED beamline in lines 69-71.

Reviewer 2 Report

Review of ‘Dosimetric optimization of a laser-driven irradiation facility 2 using the G4-ELIMED application’ by Barba et al.

The authors present a modelling study of the profiles of protons created from a laser-plasma source as would be expected for medical delivery at a system set-up at the ELI beamline facility in Prague.  The proton beam is modified by (i) ion collection and focusing optics, (ii) ion energy selection optics and (iii) transport through air to medical samples.  The authors use an ion energy spectrum as reported by Dover et al (reference 32) obtained at another laser-plasma facility and assume a Gaussian spatial spread for the protons beam originating in the laser-plasma interaction.   The work is useful and relevant to the facility development.  However, the paper is opaque and difficult to interpret mainly because of (i) a lack of detail presented on the ion beam source and optics, (ii) the use of multiple acronyms and (iii) the poor quality of the figures representing the optical components.  I suggest revising as follows.

  1. At the beginning of the paper, the authors should not assume all readers have knowledge of laser-plasma particle acceleration physics. The laser-plasma source should be discussed in more detail with an introductory paragraph.  The detail of the experimental results used for the modelling (Dover et al reference 32) could be presented (eg. type and geometry of target, irradiance on target etc).
  2. Instead of the block diagrams of figures 2 and 3, some indication of the structure of the ion optics should be presented.
  3. The authors should present more textual discussion of the ion beam optics so that their function and operation can be understood without recourse to earlier literature.
  4. The authors should justify assuming ion beam profiles that are spatially Gaussian in intensity with standard deviation of 10 microns (line 112). Similarly, the authors should clarify the assumed angular variation in figure 1b. Using X’, Y’ for an angular variation seems confusing. 
  5. Which proton beam spectrum from Dover et al is used (figure 1a)? The intensity of the laser on target for the spectrum should be given.  Why is the spectrum smooth in figure 1a, while all the published spectra in Dover et al show structure?
  6. Where does the carbon source spectrum shown in figure 5 come from? This spectrum has much more structure than the proton spectrum shown in figure 1a.

Author Response

Point 1: At the beginning of the paper, the authors should not assume all readers have knowledge of laser-plasma particle acceleration physics. The laser-plasma source should be discussed in more detail with an introductory paragraph.

Response 1: In the new version of the paper, the first paragraph of the introduction provides more details about laser-plasma ion acceleration.

Point 2: The detail of the experimental results used for the modelling (Dover et al reference 32) could be presented (eg. type and geometry of target, irradiance on target etc).

Response 2: More details about the J-KAREN-P laser were added (lines 133-135).

Point 3: The authors should explain the equation on page 9 and how to interpret it. Alternatively, they may use the proportion approach which will make the interpretation easier. Instead of the block diagrams of figures 2 and 3, some indication of the structure of the ion optics should be presented. The authors should present more textual discussion of the ion beam optics so that their function and operation can be understood without recourse to earlier literature.

Response 3: The details of the ion beam elements are presented in detail in previous papers cited in the manuscript. Nevertheless, following the Referee’s suggestion, we have added a few lines providing general information about the quadrupoles (lines 57-66) and the energy selector (Table 1 and lines 178-181).

Point 4: The authors should justify assuming ion beam profiles that are spatially Gaussian in intensity with standard deviation of 10 microns (line 112).

Response 4: We assumed a Gaussian spatial distribution with a standard deviation of 10 microns based on the simplified assumption that the proton energy spatial distribution follows the focal spot peak intensity distribution of the laser that is reported in FIG. S1b from Dover et al. (attached here).

Point 5: Similarly, the authors should clarify the assumed angular variation in figure 1b. Using X’, Y’ for an angular variation seems confusing.

Response 5: The plot b in Figure1 represents the angular aperture of the beam in the X’ Y’ phase space, as commonly described in conventional accelerators and related textbooks. In our opinion such a widely used notation does not contains ambiguity. However, according to the Referee’s comment we have changed it in θx and θy.

Point 6: Which proton beam spectrum from Dover et al is used (figure 1a)? The intensity of the laser on target for the spectrum should be given.  Why is the spectrum smooth in figure 1a, while all the published spectra in Dover et al show structure?

Response 6: We used both the angular distribution and the energy spectrum of FIG. S3 from Dover et al (see attached figure). In this case, the energy spectrum was measured at discrete energy values because it was measured with a stack of different radiochromic films. However, a real proton spectrum would contain protons with intermediate energies as well. This is the reason why we assumed such a smooth spectrum.

Point 7: Where does the carbon source spectrum shown in figure 5 come from? This spectrum has much more structure than the proton spectrum shown in figure 1a.

Response 7: It is know that in the TNSA (Target Normal Sheath Acceleration) mechanism carbon ion spectra are related with the proton spectrum as demonstrated in several tens of experimental and numerical publications in the field. Typically the C-ion energy/nucleon (energy gain in the TNSA field) is lower compared to the proton energy due to shielding effects caused by protons. This is also explained in detail in a few numerical articles, e.g. Psikal et al (Phys. Plasma 15 (2008) 053102). In our work, we artificially used a C-ion spectrum starting from the experimental proton spectrum with an energy cut-off following Equation 1. The carbon spectrum seems to have more structures compared to the proton spectrum, however this is only a visual impression due to the fact that (i) the proton was generated using 107 particles, while the carbon spectrum was generated using 105 initial particles, and (ii) the carbon spectrum is enlarged along the x axis so that small changes become more evident.

Round 2

Reviewer 2 Report

The authors have revised 'Dosimetric optimization of a laser-driven irradiation facility 2 using the G4-ELIMED application' by Barba et al following suggestions made in my first review.  The authors have not adjusted figures for the ion beam optics, but have added some further description of the ion beam optics. In addition, the authors have given a short description of the laser-plasma ion acceleration and added more details regarding the experimental results used for their simulations. The paper is suitable for publication.